# The Dual Role of Small Extracellular Vesicles in Joint Osteoarthritis: Their Global and Non-Coding Regulatory RNA Molecule-Based Pathogenic and Therapeutic Effects

**DOI:** 10.3390/biom13111606

**Published:** 2023-11-02

**Authors:** Zhi Li, Ruiye Bi, Songsong Zhu

**Affiliations:** 1State Key Laboratory of Oral Diseases & National Clinical Research Center for Oral Diseases, West China Hospital of Stomatology, Sichuan University, Chengdu 610041, China; 2022224035184@stu.scu.edu.cn; 2Department of Orthognathic and TMJ Surgery, West China Hospital of Stomatology, Sichuan University, Chengdu 610041, China

**Keywords:** small extracellular vesicles, osteoarthritis, joint, pathogenicity, therapeutic effect

## Abstract

OA is the most common joint disease that affects approximately 7% of the global population. Current treatment methods mainly relieve its symptoms with limited repairing effect on joint destructions, which ultimately contributes to the high morbidity rate of OA. Stem cell treatment is a potential regenerative medical therapy for joint repair in OA, but the uncertainty in differentiation direction and immunogenicity limits its clinical usage. Small extracellular vesicles (sEVs), the by-products secreted by stem cells, show similar efficacy levels but have safer regenerative repair effect without potential adverse outcomes, and have recently drawn attention from the broader research community. A series of research works and reviews have been performed in the last decade, providing references for the application of various exogenous therapeutic sEVs for treating OA. However, the clinical potential of target intervention involving endogenous pathogenic sEVs in the treatment of OA is still under-explored and under-discussed. In this review, and for the first time, we emphasize the dual role of sEVs in OA and explain the effects of sEVs on various joint tissues from both the pathogenic and therapeutic aspects. Our aim is to provide a reference for future research in the field.

## 1. Introduction

Osteoarthritis (OA) is the most common joint disease that is highly related to age and mechanical factors. As a chronic degenerative disease, it mainly damages the articular cartilage and destroys the entire joint articular surface [1]. Progressive softening, abrasion and thinning of the articular cartilage induce typical lesions, and the main symptoms of OA include joint pain, stiffness and swelling [2]. This disease occurs in almost all joints, especially in the weight-bearing joints such as the hip, knee and spine. According to the WHO Global Burden of Disease Study, the lifetime risk of morbidity associated with hip OA is approximately 25%, which increases to 45% for knee OA [2].

Due to the high morbidity rate, the life quality of OA patients is seriously affected, and it imposes a heavy economic burden on families and society [3,4,5]. There are reports that the direct economic losses caused by OA each year account for 1–2.5% of the GDP in Canada and America [2,4], while the indirect losses due to the reduction in social production activities affected by the health status of patients are difficult to assess. In addition, the incidence of OA is highly correlated with age, and in countries with increasingly aging populations, these losses will increase year by year [5,6].

However, due to the lack of understanding of the pathogenesis of OA, there is no ideal therapeutic approach to cure or relieve this disease. Currently, the treatment of early OA includes the use of nonsteroidal anti-inflammatory drugs (NSAIDs) and pain relievers, as well as nonpharmacological treatments such as exercise, weight loss and physical therapy [1,7]. However, these methods can only produce temporary relief and fail to repair damaged articular cartilage, and they also present with problems such as poor long-term treatment effect and uncertain safety. For advanced patients with severe dysfunction, surgical intervention is required such as total joint replacement, but complications such as infection and implant loosening may occur, which result in poor functional recovery and eventual disability [1,8].

Almost all cells can secrete extracellular vesicles, and those with a diameter of <100 nm or <200 nm are named small extracellular vesicles (sEVs) [9]. These vesicles have a phospholipid bilayer structure with special surface molecules, and contain different cargoes such as nucleic acids, proteins and metabolites [9,10]. They can cross the cytoplasmic membrane, the blood–brain barrier and the gastrointestinal tract, and can be taken up by recipient cells, thus playing an important role in cellular communication [11].

sEV-based therapy is considered a potential treatment strategy for OA. It has attracted wide attention with the key role of sEVs in stem cell therapy being revealed in recent years [12,13,14,15,16,17,18,19,20,21,22,23,24]. Compared to stem cell treatments, treatments with sEVs show similar efficacy [25] while providing safer therapeutic effect without potential adverse outcomes, such as uncertain differentiation direction and immunogenicity [3]. On the one hand, sEVs not only have therapeutic functions but also pathogenic roles. For example, Gao [26] et al. found that sEVs in the synovial fluid (SF) of OA patients inhibited the proliferation of chondrocytes. On the other hand, sEVs can be endogenous or exogenous, derived from various cells, and used alone or in combination with tissue engineering, and the mechanism becomes gradually clearer with the effect of different specific molecular mechanisms being revealed. The miRNAs (microRNAs), lncRNAs (long non-coding RNAs) and circRNAs (circular RNAs) carried by sEVs produce effect via activating or inhibiting various signaling pathways.

This review summarizes previous research results and classifies them from the perspective of the effects of sEVs on the pathogenesis and treatment of OA. The aim of this review is to provide a reference for further research.

## 2. Origin of sEV Therapy: Stem Cell Therapy for Osteoarthritis

The damage and death of chondrocytes is one of the key lesions in OA [1], which suggests a potential therapeutic strategy for OA. Autologous chondrocyte implantation (ACI) at the lesion cartilage is considered the most traditional cellular therapy for OA [27]. It has the advantages of a high success rate and good safety, and the problem of residency rate and dedifferentiation can also be improved through the combined application of various cell scaffolds [28]. However, its efficacy is ultimately limited by the insufficient numbers of autologous chondrocytes available. Inducing stem cell differentiation into chondrocytes to fill the lesion cartilage in vivo is a possible solution because stem cells are more accessible. Intra-articular injection or surgical implantation of MSCs derived from different tissue sources, such as bone marrow, adipose and umbilical cord, has been reported to have regenerative repair effects on OA knees [29,30,31,32]. In temporomandibular joint (TMJ) OA, it has also been reported that the defect is repaired after intra-articular injection of MSCs [33,34,35,36,37,38,39,40]. Moreover, compared to BMSCs, tooth-derived stem cells may display a higher affinity for regenerating craniofacial tissues because they share the same embryological origins and similar gene expression patterns [35,41,42].

Although stem cell therapy has experienced further progress, compared to ACI, some new problems have arisen, mainly due to the strong tumorigenic potential and immunogenicity of stem cells [43]. Investigators have found that the regenerative effects of stem cell therapy are partly due to the paracrine effects of stem cells, in addition to their chondrogenic differentiation. Specifically, stem cells in the body can secrete information substances, which do not enter the blood circulation but affect their adjacent cells through diffusion. These information substances include growth factors, nitric oxide and hormones. They regulate downstream gene expression in cells, which implies higher safety, predictability and controllability for the final effects. These advantages have attracted the attention of researchers. With a deeper insight obtained from further research, some investigators have even emphasized that the role played by the paracrine effects in OA is actually greater than previously believed, though more evidence is needed. sEVs, as the main bearers of the paracrine effects, have received increasing attention in this field.

## 3. Effects of sEVs on Articular Tissues in OA

During the OA process, sEVs exhibit either a pathogenic or therapeutic effect. Pathogenic sEVs cause or promote lesions and are commonly derived from adjacent pathological tissues of the joint. Therapeutic sEVs have an opposite effect, and they are commonly derived from healthy tissues throughout the body or are injected after extraction to work in the body [44,45,46]. At the same time, the main regions that are affected in the synovial joint include chondrocytes, the cartilage matrix, the subchondral bone and the synovium, which can produce different effects under the action of the same sEVs.

Hereafter, sEVs that meet at least one of the following criteria are classified as therapeutic sEVs: a) they have been found to exert therapeutic effects on OA in in vivo experiments, such as cartilage regeneration and pain relief, and b) they have been found to have at least one significant anti-OA effect in in vitro experiments, including promoting chondrocyte proliferation and migration, or inhibiting their apoptosis; promoting cartilage matrix secretion or inhibiting their degradation; promoting synovial inflammation resolution; and promoting subchondral bone recovery. sEVs with the opposite effects are classified as pathogenic sEVs [47].

### 3.1. Pathogenic sEVs in OA

#### 3.1.1. Effects of Pathogenic sEVS on Chondrocytes

Chondrocytes are the main damaged objects in OA, and their affected biological behaviors include proliferation, hypertrophy, apoptosis, senescence, differentiation, migration, inflammatory response and metabolism. SEVs that affect the progress of OA play a pathogenic role in OA chondrocytes, and these sEVs are derived from (a) synovial fibroblasts (SFBs) in synovium [26,48,49,50,51]; (b) osteoblasts and osteoclasts in the subchondral bone [52,53]; and (c) vascular endothelial cells in nearby vessels [54].

Gao [26] et al. reported increased inflammatory cytokines and chemokines in sEVs derived from the synovial fluid (SF) of patients during OA progress, leading to the inhibition of chondrocyte proliferation. Further studies [48,49,50] revealed that pathogenic sEVs in the SF were derived partly from SFBs with OA-like lesions. In addition to inhibiting chondrocyte proliferation, these SFB-sEVs were found to inhibit migration, induce apoptosis and promote the inflammatory response of chondrocytes, according to the studies by Zhou [48] et al. and Kato [51] et al. 

At the same time, Wu [52] et al. and Dai [53] et al. separately showed that osteoblast-derived sEVs and osteoclast-derived sEVs from the subchondral bone could promote hypertrophy and catabolism of articular chondrocytes in OA. Yang [54] et al. found that sEVs from vascular endothelial cells reduced the resistance of chondrocytes to oxidative stress by inhibiting autophagy and p21 expression, thereby inducing chondrocyte apoptosis. 

#### 3.1.2. Effects of Pathogenic sEVS on Cartilage Extracellular Matrix

The cartilage extracellular matrix (ECM) is a complex network composed of various macromolecules secreted by chondrocytes [1]. The expression levels of catabolic markers, such as MMP13, ADAMTS4 and ADAMTS5, in chondrocytes are positively correlated with the degree of ECM destruction, while the expressions of anabolic markers, such as COL2A1 and ACAN, have a negative correlation with intensified ECM destruction.

sEVs derived from OA hCCs, SFBs and osteoblasts play a pathogenic role in the cartilage ECM. Guo [55], Zhu [56], Lai [57] and Meng [58] et al. found that hCC-sEVs in OA aggravate the up-regulation of MMP13 and down-regulation of ACAN in chondrocytes, ultimately causing cartilage ECM degradation. sEVs derived from OA SFBs have been shown to promote the degradation of OA cartilage ECM by sequestering miR-142-5p and upregulating RUNX2 [59]. In the study by Wu [52] et al., sEVs derived from the osteoblasts from the sclerotic subchondral bone of OA patients triggered catabolic gene expression in chondrocytes.

In addition, some sEVs derived from healthy cells have been shown to pro-degrade cartilage ECM as a side effect. There are several investigations that demonstrate that normal hS-MSCs [13] and human urine-derived stem cells (hUSCs) [60] further reduce cartilage ECM secretion in OA chondrocytes.

#### 3.1.3. Effects of Pathogenic sEVS on Synovial Tissue

The synovium is a thin and smooth layer of loose connective tissue located in the inner layer of the joint capsule. Its inflammation is one of the typical lesions in OA, whose severity degree is determined by the types and levels of cytokines. And these cytokines are secreted by SFBs and synovial macrophages.

According to the existing findings, the destruction of synovial tissue by OA sEVs is mainly through internally enriched pro-inflammatory factors, including IL-1β, IL-6, IL-12, IL-18, TNF-α and NF-κB. However, sEVs derived from OA hCCs are currently the only reported pathogenic sEVs. Ni [61] et al. found that the production of mature IL-1β in macrophages is enhanced by sEVs and subsequently exerts a pro-OA effect, which involves the inhibition of macrophage autophagy.

### 3.2. Therapeutic sEVs in OA

#### 3.2.1. Effects of Therapeutic sEVS on Chondrocytes

Mesenchymal stem cells (MSCs) have been shown to secrete therapeutic sEVs to maintain the function of chondrocytes in OA joints. Wang [62] et al. and Liu [14,63] et al. found that normal murine bone marrow MSC-sEVs (BMSCs-sEVs) and human bone marrow MSC-sEVs could enhance the proliferation of chondrocytes in OA [64]; meanwhile, hBMSC-sEVs inhibited chondrocyte hypertrophy. The ability to inhibit chondrocyte apoptosis in OA was also observed in sEVs derived from BMSCs [13,18,33,35,36,37,38,39,40], human umbilical cord MSCs [65] (hUC-MSCs), dental pulp stem cells [18] (DPSCs), human synovial MSCs [17,66] (hS-MSCs) and infrapatellar fat pad (IPFP)-MSCs [21]. And sEVs derived from BMSCs [19], UC-MSCs [67,68] and hypoxia-cultured human adipose-derived stem cells [69] have been reported to be able to restore aging chondrocytes. In terms of inflammatory response, BMSCs-sEVs [15] and DPSCs [70] could inhibit the production of inflammatory factors, such as IL-1, IL-6 and TNF-α. In addition, the promotion of chondrogenic differentiation driven by MSCs-sEVs has been reported by Zeng Li [71] and Guping Mao [12] et al.

Chondrocyte-derived sEVs(hCC-sEVs) also play a protective role for cartilage in OA. Li [71] et al. and Ma [72] et al. used hCC-sEVs to treat human bone marrow MSCs (hBMSCs) and hUC-MSCs, respectively. Both studies found that the transcriptional activities of chondrogenesis, such as ACAN, COL2A1 and SOX9 expression levels, were significantly increased.

In addition to being used alone, sEVs can also be applied to a lesion area in combination with gels [73] or tissue engineering constructs [74]. Apart from improving the drug retention rate, sEVs can also provide differentiation signals when used in conjunction with tissue engineering constructs. hBMSC-sEVs and hCCs-sEVs have been reported to provide chondrogenic signals for rabbit chondrogenic progenitor cell constructs. Compared to hBMSC-sEVs, the cartilage tissue induced by hCCs-sEVs has a lower degree of cell hypertrophy and almost no vascularization, which is more consistent with physiological cartilage [74].

#### 3.2.2. Effects of Therapeutic sEVS on Cartilage Extracellular Matrix (ECM)

Current research evidence shows that therapeutic sEVs in the cartilage ECM are mainly derived from various MSCs. Among several EVs which are reported to have the effect of promoting cartilage ECM deposition, BMSCs-sEVs are the most studied at present [75,76,77,78,79,80,81,82], whose effect can be enhanced by low-intensity pulsed ultrasound (LIPUS) [83,84] and parathyroid hormone (PTH) pretreatment [85]. In addition, sEVs derived from human embryonic MSCs (hE-MSCs) [21,86], human adipose MSCs(hA-MSCs) [87], PDLSCs (periodontal ligament-derived stem cells) [88] and monocyte-derived cells [89] have been shown to affect the expression of related genes that are involved in COL2A1, ACAN, MMP13, ADAMTS4 and ADAMTS5, thereby promoting the deposition of cartilage ECM.

#### 3.2.3. Effects of Therapeutic sEVS on Subchondral Bone

The subchondral bone maintains the morphology of the joint, supplies nutrition to the cartilage, and bears the joint load. Current studies have mainly focused on the effect of therapeutic sEVs on subchondral bone metabolism during OA. Generally speaking, sEVs are able to restore abnormal bone index such as bone volume fraction (bone volume/total volume) (BV/TV) and trabecular spacing (Tb.Sp.), as well as increasing osteoblast number and decreasing osteoclast number.

In a rat knee OA model, Zhang [21] et al. and Jin [19,90] et al. observed that intra-articular injection of hE-MSCs-sEVs or hBMSCs-sEVs significantly restored the BV/TV and Tn.Sp, which was accompanied by a significant increase in osteoblast number and decrease in osteoclast number. In addition to the restoration of subchondral bone structure, therapeutic sEVs can regulate the growth of nerves and blood vessels in the subchondral bone to provide relief of joint pain [91]. Li [92] and Wang [93] et al. observed that, in the lumbar facet joints and knee joints of mice, BMSCs-sEVs significantly alleviated pain by inhibiting abnormal CGRP-positive nerve invasion and abnormal H-type blood vessel formation, respectively, in OA subchondral bone remodeling. Li [92] et al. further revealed that this effect may be mediated through the RANKL-RANK-TRAF6 signaling pathway in the subchondral bone. Considering that the RANKL/RANK system plays an important role in bone remodeling, further exploration in this direction is of great significance for revealing the mechanism of subchondral bone remodeling in OA.

#### 3.2.4. Effects of Therapeutic sEVS on Synovial Tissue

Regarding SFBs, sEVs derived from normal human adipose-derived stem cells (hADSCs) and hBMSCs can down-regulate the expression of IL-6, NF-κB and TNF-α, as well as up-regulate the expression of IL-10 in OA-hSFBs, ultimately improving synovial inflammation [94,95,96].

Regarding synovial macrophages, it has been shown that rat BMSCs-sEVs can promote the transformation of macrophages from M1 to M2 type, thereby promoting the secretion of IL-10 but inhibiting IL-1β and TNF-α via synovial macrophages [62,97]. In addition, co-culturing sEV-treated M2-polarized macrophages with chondrocytes also helps maintain the cartilage characteristics of chondrocytes and inhibit their hypertrophy and senescence [97,98] (Figure 1).

## 4. Direct Bearers of Sev Effects on OA: Non-coding Regulatory RNA Molecule

The species of the cell source, the different cell types and the physiological/pathological condition of cells merely indirectly determine the function of sEVs; it is the sEV cargoes that really exert a direct determining effect and are the direct bearers of the effects of sEVs in OA.

sEV cargoes mainly include three types of non-coding regulatory RNA molecules: (a) microRNAs, (b) long non-coding RNAs (lncRNAs) and (c) circular RNAs (circRNAs). Each specific cargo could have specific effects (Figure 2).

### 4.1. MicroRNAs (miRNAs)

Mechanically, miRNAs bind to the 3’-untranslated region (3’-UTR) of target mRNAs to inhibit their translation, and then negatively regulate the expressions of substream proteins, such as mTOR, NF-κB and WNT5A, leading to the promotion or inhibition of OA [99]. As the most abundant sEV cargoes, miRNAs are generally considered to have the most important role in sEV effects. Thus, there has been a large number of research studies investigating their roles and mechanisms in the OA process (Table 1.).

The NF-κB signaling pathway is one of the signaling pathways activated in OA. It regulates the production of many inflammatory mediators and cytokines, and participates in many events such as chondrocyte apoptosis and catabolism, synovial inflammation and chondrogenic differentiation. This pathway has long been regarded as a potential target for OA treatment [100]. Studies examining the treatment of OA with sEVs have revealed that the NF-κB signaling pathway may be involved in the anti-OA effect of sEVs. Related studies have reported that it is mainly the different miRNAs carried by sEVs that ultimately cause the NF-κB signaling pathway to be inhibited. In chondrocytes, both miR-129-5p and miR-93-5p have been reported to increase cell viability and inhibit apoptosis, immune response and catabolism through the HMGB1/TLRs4/NF-κB axis [17,58]; miR-326 exerts similar effects through the HDAC3/STAT1/NF-κB axis [101]. The inhibition of chondrocyte apoptosis by sEV miR-143 has also been reported to be related to the inhibition of the NF-κB signaling pathway [102]. Several studies have shown that sEV-mediated inhibition of the NF-κB pathway may be achieved by inhibiting IκBα degradation [81,83,96]. Usually, IκBα combines with NF-κB to form an intracytoplasmic complex, thereby preventing NF-κB from entering the nucleus to play its role. In the OA state, through the mediation of intrinsic membrane receptors, some extracellular signaling substances can activate an enzyme called IkB kinase (IKK). Then, IKK phosphorylates IκBα protein, which leads to the ubiquitination of the latter, causing IκBα to detach from NF-κB, and, eventually, IκBα is degraded by proteasome [100]. hBMSCs-sEVs have been reported to target DDX20 by delivering miR-361-5p, thereby causing an attenuation of IκBα degradation and inhibiting the activation of the NF-κB signaling pathway in human chondrocytes [81]. However, a single application of SD rat BMSCs-sEVs was reported to not cause changes in NF-κB signal intensity in rat chondrocytes, although it still exerted an OA inhibitory effect. Interestingly, when combined with LIPUS, inhibition of IκBα degradation was observed in chondrocytes, and the therapeutic effect of BMSCs-sEVs in SD rats with OA was enhanced [83]. In human synovial fibroblasts, sEV miR-147b, miR-145, and miR-221 were reported to inhibit the activation of the NF-κB signaling pathway [94,96]. And the latter two miRNAs were also reported to exert anti-OA effects by promoting the conversion of synovial macrophages from M1 to M2 phenotype and stimulating the chondrogenic differentiation of periosteal cells [94].

**Table 1 biomolecules-13-01606-t001:** MicroRNAs of mechanistic importance in OA.

MicroRNA	Origin of sEV	Mechanism	Effects	Species	Ref.
miR-9-5p	rat BM-MSCs	Negatively regulates SDC1	Reduces inflammation in OA cartilage (decreases IL-1, IL-6, TNF-α, CRP, NO, MDA, iNOS, COX2, SOD, OCN, MMP13, COMP, AKP)	rat	[15]
miR-92a-3p	human BM-MSCs	Negatively regulates WNT5A	Increases cell proliferation and matrix secretion in OA cartilage (increases ACAN, COL2A1, COL9A1, COMP, SOX9; decreases COL10A1, RUNX2, MMP13)	human	[12]
miR-95-5p	human BM-MSCs	Negatively regulates HDAC2/8	Increases cartilage development and cartilage matrix expression in MSCs and chondrocytes (increases ACAN, COL2A1, COL9A1, COMP; decreases COL10A1, MMP13)	human	[79]
miR-100-5p	human UC-MSCs	Negatively regulates NOS4	Reduces ROS production and apoptosis in OA chondrocytes	human	[65]
miR-100-5p	human IPFP-MSCs	Negatively regulates mTOR	Increases autophagy and reduces apoptosis in OA chondrocytes; increases secretion in OA ECM (increases COL2; decreases MMP13, ADAMTS5)	human	[21]
miR-100-5p	human exfoliated deciduous teeth-SCs	Negatively regulates mTOR	Reduces inflammation in OA chondrocytes (decreases MMP1, MMP9, MMP13, ADAMTS5)	human	[70]
miR-124	MSCs	Negatively regulates ROCK1, which actives TLR9	Reduces apoptosis in OA chondrocytes	mouse	[102]
miR-125a-5p	human BM-MSCs	Negatively regulates E2F2	Increases migration in OA chondrocytes and increases secretion in OA ECM (increases COL2, ACAN, SOX9; decreases MMP13)	human/mouse	[75]
miR-126-3p	rat SFBs		Increases migration and proliferation, reduces apoptosis and inflammation in OA chondrocytes (decreases IL-1β, IL-6, TNF-α); reduces formation of osteophytes and degeneration in cartilage	rat	[48]
miR-127-3p	rat BM-MSCs	Negatively regulates CDH11, which actives Wnt/β-catenin pathway	Increases cell viability and DNA synthesis activity, and reduces apoptosis in OA chondrocytes (increases COL2; decreases MMP13)	rat	[103]
miR-129-5p	human S-MSCs	Negatively regulates HMGB1, which up-regulates TLR4 and then actives NF-κB signaling pathway	Reduces inflammatory response and apoptosis in OA chondrocytes (increases COL2; decreases COX2, iNOS, MMP13)	human	[17]
miR-135b	rat MSCs	Negatively regulates Sp1	Increases viability and proliferation in OA chondrocytes	rat	[64]
miR-135b	rat BM-MSCs	Negatively regulates MAPK6	Increases M2 polarization in synovial macrophages; increases repair of OA cartilage (increases ACAN, SOX9); reduces inflammatory factors (IL-1β, PGE2, COX-2, COX-1, NO) in OA serum	rat	[62]
miR-136-5p	human BM-MSCs	Negatively regulates ELF3	Increases migration in chondrocytes; reduces degeneration in cartilage (increases COL2, ACAN, SOX9; decreases MMP13)	human/mouse	[76]
miR-140-5p	human DPSCs/human USCs	Negatively regulates VEGFA	Increases proliferation, migration, and reduces apoptosis in OA chondrocytes; increases secretion in OA ECM (increases COL2, ACAN, SOX9)	human/rat	[18,60]
miR-140-5p	human S-MSCs	Negatively regulates RalA	Increases secretion in OA ECM (increases SOX9, ACAN, and COL2)	human/rat	[13]
miR-143	MSCs	Negatively regulates ROCK1, which actives NF-κB signaling pathway	Reduces apoptosis in OA chondrocytes	mouse	[102]
miR-145	human ADSCs	inhibits the NF-κB signaling pathway; actives Wnt/β-catenin signaling pathway	Increases chondrogenesis (increases SOX9 in periosteal cells) and collagen deposits in OA cartilage	human	[94]
miR-147b	human BM-MSCs	inhibits the degradation of IκBα, then inhibits the NF-κB signaling pathway	Reduces inflammation in OA synovial cells (increases SOCS3, SOCS6; decreases IL-1β, IL-6, monocyte chemoattractant protein-1)	human	[96]
miR-155-5p	human S-MSCs	Negatively regulates Runx2	Increases proliferation, migration, ECM secretion (increases COL2, SOX9) and reduced apoptosis in OA chondrocytes	human/mouse	[66]
miR-206	mouse	Negatively regulates ELF3	Increases proliferation, differentiation and reduces apoptosis in OA osteoblasts (increases OCN, BMP2, ALP and calcium deposition level; decreases pro-inflammatory mediators)	mouse	[90]
miR-210-5p	human subchondral bone osteoblast	May involve PI3K/AKT/mTOR signaling pathway	Increases hypertrophic, degradative gene expression, and changes cellular aerobic respiration in OA chondrocytes (increases ADAMTS5, COL10, RUNX2, MMP13; decreases SOX9, ACAN, COL2)	human	[52]
miR-214-3p	rat SFBs		Increases proliferation, and reduces apoptosis, inflammation (decreases TNF-α, IL-1β) in OA chondrocytes; reduces degeneration of OA cartilage and synovium; reduces osteophytes formation and maintains the subchondral bone structure in OA	rat	[49]
miR-221	human ADSCs	inhibits the NF-κB signaling pathway; actives Wnt/β-catenin signaling pathway	Increases viability and proliferation of periosteal cells	human	[94]
miR-320c	S-MSCs	Negatively regulates ADAM19, which up-regulates β-catenin, MYC and then actives Wnt signaling pathway	Increases proliferation, migration and reduces apoptosis in OA chondrocytes; increases secretion in OA ECM (increases COL2A1, ACAN)	rat	[104,105]
miR-320c	human BM-MSCs		Increases proliferation in OA chondrocytes; increases secretion in OA ECM (increases SOX9; decreases MMP13)	human	[80]
miR-326	rat BM-MSCs	Negatively regulates HDAC3, which negatively regulates STAT1 and then actives NF-κB signaling pathway	Increases proliferation, migration, the expression of chondrogenic specific genes (COL2A1, SOX9, Agg, and Prg4), and reduce inflammation(IL-6, TNF-α), pyroptosis (decreases NLRP3, ASC, GSDMD, Caspase-1, IL-1β, IL-18) in OA chondrocytes	rat	[101]
miR-361-5p	human BM-MSCs	Negatively regulates DDX20, which actives NF-κB signaling pathway	Reduces inflammation (decreases IL-18, IL-6, TNF-α) and degradation (decreases iNOS, MMP3, MMP13) in OA cartilage	rat	[81]
miR-let-7a-5p	mouse osteoclast	Negatively regulates Smad2	Increases hypertrophic gene expression in OA chondrocytes (increases COL10A1, RUNX2, MMP13)	mouse	[53]

### 4.2. Long Non-Coding RNAs (lncRNAs)

lncRNAs act as competing endogenous RNAs (ceRNAs) to separate miRNAs from target mRNAs [106]. Although there are many reports on the effects of lncRNAs on chondrocytes, there are only a few studies on sEV lncRNAs [107] (Table 2).

### 4.3. Circular RNAs (CircRNAs)

CircRNAs belong to a special class of non-coding RNA molecules with a closed-loop structure. They are the same ceRNAs as lncRNAs and act as miRNA sponges with numerous miRNA binding sites. By interacting with OA-associated miRNAs, circRNAs play an important regulatory role in OA [108] (Table 3).

## 5. Limitation and Future Direction

### 5.1. Limitations in the Mechanism Research for OA Diagnosis and Treatment

Osteoarthritis can be divided into different disease stages, and the changes in sEVs in each stage are different. Moreover, among sEVs secreted by different cells, their changes in the same stage are also different. For diagnostic applications of sEVs, further research is still required to supplement current data, especially data on the early stage of OA. The data that have been reported thus far all focus on the progressive stage, and are not enough to provide a reference for establishing the criteria for early diagnosis of OA. Fortunately, the advancement in high-throughput detection technologies has made it easier to collect these data, but the participation of more investigators is needed.

Regarding treatment, the search for MSC subpopulations capable of producing more sEVs is an important issue to be solved in the clinical application of sEVs. Bone marrow, umbilical cord, antler and ginger have all shown to be potentially ideal sources. In addition, the artificial modification of MSCs is another feasible approach. The earliest attempt to continuously obtain sEVs by immortalizing MSCs was carried out by Chen et al. [110]. However, these immortalized cells, which accept lentiviral transfection of the MYC gene, largely lose their multilineage differentiation. Fortunately, the sEVs secreted by them are still effective in the treatment of myocardial infarction and OA, and are secreted in significantly higher amounts than in normal cells [24]. Currently, more immortalization protocols for MSCs have been validated, and some can still preserve good multiple differentiated potentiality, such as SV40LT and hTERT [111], but the quantity and functional changes of the sEVs secreted by these cells lack further reports. As genetic engineering technology matures, it becomes possible to directly regulate genes involved in sEV production.

In fact, in recent years, it has been reported that the amount of sEV miRNAs is not sufficient to cause an effect on target cells [112,113]. This disputes the hypothesis in previous studies that the observed regulatory effects are caused by sEVs delivering miRNAs into target cells. On the one hand, this suggests that sEV cargoes may not play a direct role, and there may be more hidden steps between miRNAs and mRNAs. On the other hand, it also suggests that what actually functions may not be sEV miRNAs, or even any sEV components [114]. This consideration stems from the shortcomings of current technology of sEV isolation and extraction. The final obtained sEV sample is actually a mixture, and it is still unclear whether unknown components or the interactions of different components affect the final effect [115,116,117]. In addition, the transfection technology used in sEV research can overexpress specific miRNAs that are several orders of magnitude above physiological levels [118], which is reported to exert functions that sEVs do not have under physiological conditions [119].

### 5.2. Limitations in sEV Isolation Technology

Current sEV extraction methods cannot guarantee yield and purity simultaneously. In terms of diagnosis, whether sEVs isolated and used for detection reflect the real situation in an organism remains to be determined. The influence of sEV depletion during extraction and the presence of certain impurities, which are hard to remove, on subsequent analysis are unclear [115,116,117]. Regarding treatment, even though the most common therapeutic sEVs are currently derived from MSCs, a kind of cells that secrete more sEVs than other cells, their production still fails to meet clinical needs. And the effects of impurities are unknown. Perhaps, developing and improving assays for pooled samples is a promising way for improving the current status.

## Figures and Tables

**Figure 1 biomolecules-13-01606-f001:**
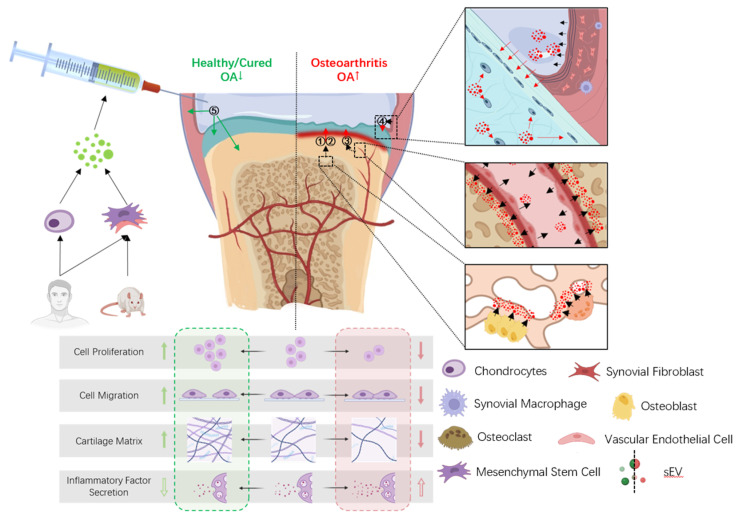
Pathogenic sEVs and therapeutic sEVs in OA. In OA joints, osteoblasts (①), osteoclasts (②), vascular endothelial cells (③), synovial fibroblasts and macrophages (④) secrete (black arrow) sEVs (red) that act on the cartilage and produce pathogenic effects (red arrow). Healthy human chondrocytes, MSCs and rat MSCs (⑤) can secrete sEVs (green) that act on the cartilage, synovium and subchondral bone and produce therapeutic effects (green arrow).

**Figure 2 biomolecules-13-01606-f002:**
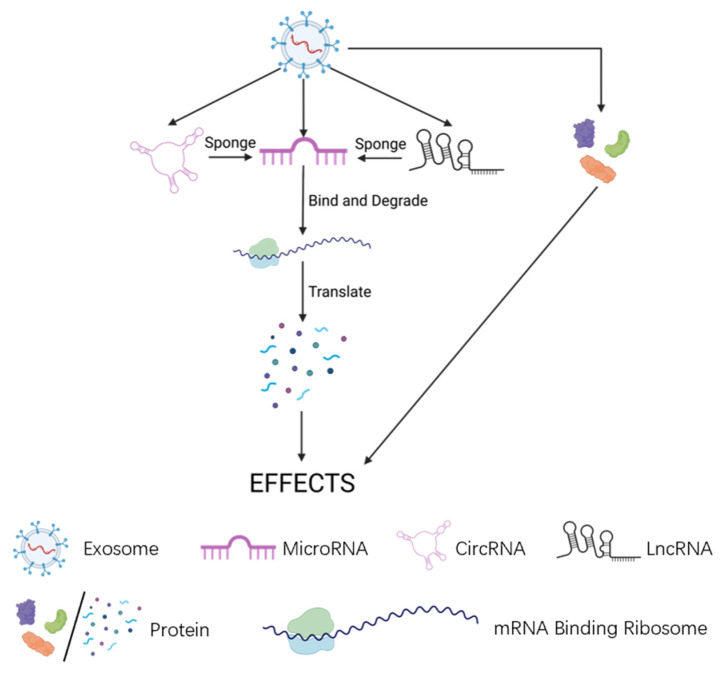
Mechanism of sEVs cargoes.

**Table 2 biomolecules-13-01606-t002:** LncRNAs of mechanistic importance in OA.

LncRNA	Origin of sEV	Mechanism	Effects	Species	Ref.
KLF3-AS1	human MSCs	Sponges miR- 206, which negatively regulates GIT1	Increases proliferation and reduces apoptosis, hypertrophy (decreases MMP13, Runx2) in OA chondrocytes, and increases secretion in OA ECM (increases COL2A1, ACAN)	mouse	[63]
PCGEM1	human SFBs	Sponges miR-142-5p, which negatively regulates RUNX2	Increases apoptosis in OA chondrocytes; increases degradation in OA ECM (increases MMP13; decreases ACAN, COL2A1)	human	[59]
PVT1	human CCs	Sponges miR-93-5p, which negatively regulates HMGB1 and then negatively regulates TLR4 to inhibit NF-κB signaling pathway	Increases apoptosis (increases Bax and cleaved caspase-3; decreases Bcl-2), inflammation responses (increases IL-6, IL-1β, TNF-α) and reduces viability in OA chondrocytes; increases collagen degradation in OA ECM(increases MMP13; decreases ACAN)	human	[58]
MEG-3	human BM-MSCs	May involve miR-206/GIT1, miR-92a-3p/Wnt5a, miR-93/TGFBR2, miR-16/SMAD7 axis	Reduces senescence and apoptosis in OA chondrocytes	rat	[19]
H19	human UC-MSCs	Sponges miR-29a-3p, which negatively regulates FOS	Reduces pain and central sensitization in advanced OA	rat	[91]
	human UC-MSCs	Sponges miR-29a-3p, which negatively regulates FOXO3	Increases migration and reduces senescence and apoptosis in OA chondrocytes; increases secretion in OA ECM (increases COL2A1, ACAN)	human/rat	[68]
	rat SFBs	Sponges miR-106b-5p, which negatively regulates TIMP2	Increases proliferation and migration in OA chondrocytes; reduces degeneration in OA cartilage (increases COL2A1, ACAN; decreases MMP13, ADAMTS5)	human	[50]

**Table 3 biomolecules-13-01606-t003:** CircRNAs of mechanistic importance in OA.

CircRNA	Origin of sEV	Mechanism	Effects	Species	Ref.
circ_0001236	human BM-MSCs	Sponges miR-3677-3p, which negatively regulates Sox9	Regulates degradation and repair in OA ECM (increases COL2A1 and SOX9; decreases MMP13)	mouse	[82]
circ_0001846	human CCs	Sponges miR-149-5p, which negatively regulates WNT5B	Reduces cell viability, invasion and migration, and increases apoptosis, inflammatory cytokines production in OA chondrocytes; and increases degradation in ECM	human	[56]
circBRWD1	human CCs	Sponges miR-1277, which negatively regulates TRAF6	Reduces cell viability and proliferation and increases apoptosis (increases Bax; decreases CyclinD1), inflammation (increases IL-6, IL-8) and ECM degradation (increases MMP13; decreases ACAN) in OA chondrocytes	human	[55]
circCDK14	human CCs	Sponges miR-1183, which negatively regulates KLF5	Increases proliferation, and reduces apoptosis in OA chondrocytes; reduces degradation in OA ECM	human	[57]
circPRKCH	human CCs	Sponges miR-502-5p, which negatively regulates ADAMTS5	Reduces proliferation, migration, and increases apoptosis, inflammatory response to promote phenotypic changes in OA chondrocytes	human	[109]

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
