# Peer review of "The Dual Role of Small Extracellular Vesicles in Joint Osteoarthritis: Their Global and Non-Coding Regulatory RNA Molecule-Based Pathogenic and Therapeutic Effects"

_biomolecules, 2023, doi:10.3390/biom13111606_

Round 1

Reviewer 1 Report

Please revise extensively for the English language

115-118 : promoting proliferation, and cell migration, and  gene expression, are not suppressing effects. Add a proper reference

125-127. Please rewrite the sentence. “Which” is referred, I guess, to EOS, but in the phrase it seems to refer to Chondrocytes

141 substitute “derive” with a more appropriate terminology

147  a space is missing: or infrapatellar

272 "stages is still awaiting supplementary, especially at the early stage". ….are awaiting.

The sentence is not clear

3.1.1; 3.1.2;3.2.1, 3.2.2; 3.3.1 , ; 3.3.2; 3.4.1  and 3.4.2

The discussion of the revision would be of greater interest if enriched by tables as have been included in sections 4.1-4.2-4.3-4.4

This revision would be of greater interest if enriched by tables as have been included in sections 4.1-4.2-4.3-4.4

Several grammatical and syntax errors. Some words are not appropriate. Unclear sentences.

Author Response

Dear Editorial Board of Biomolecules,

We really appreciate the chance from Biomolecules for revising our manuscript. To address the questions from the editors and reviewers, we have carefully revised the ‘Introduction’, ‘Effects of sEVs on articular tissues in OA’, ‘Direct bearers of sEV effects on OA: Non-coding Regulatory RNA Molecule’ and ‘Limitations in the mechanism research for OA diagnosis and treatment’ sections accordingly.

Reviewers' comments:

Reviewer 1:

  1. 115-118 : promoting proliferation, and cell migration, and gene expression, are not suppressing effects. Add a proper reference.

We really appreciate your kind comment about our work. And we are sorry for any confusion caused by using improper words. The real meaning of “OA suppressing effect” in the manuscript is “anti-OA effect”. Therefore, we change “have at least one significant OA suppressing effect” to “have at least one significant anti-OA effect”. And we add a proper reference here.

  1. 125-127. Please rewrite the sentence. “Which” is referred, I guess, to EOS, but in the phrase it seems to refer to Chondrocytes.

We change “which” to “and these sEVs are”.

  1. 141 substitute “derive” with a more appropriate terminology.

We change “derive” to “secrete”.

  1. 147 a space is missing: or infrapatellar.

Thank you very much for your careful review. We have corrected this clerical error.

  1. 272 "stages is still awaiting supplementary, especially at the early stage". ….are awaiting. The sentence is not clear.

We rewrite “On diagnosis, the different expression patterns of different sEVs at different OA stages are still awaiting supplementary, especially at the early stage” to “Osteoarthritis can be divided into different disease stages, and the changes in sEVs in each stage are different. Moreover, among sEVs secreted by different cells, their changes in the same stage are also different. For diagnostic applications of sEVs, further research is still required to supplement current data, especially data on the early stage of OA”. And thank you again for your careful review, we have corrected this clerical error.

  1. 1.1; 3.1.2;3.2.1, 3.2.2; 3.3.1; 3.3.2; 3.4.1 and 3.4.2. The discussion of the revision would be of greater interest if enriched by tables as have been included in sections 4.1-4.2-4.3-4.4.

This is a very important comment. Following your comments, we enrich the table in “4. Direct bearers of sEV effects on OA” section with relevant data in “3. Effects of sEVs on articular tissues in OA” section. These mainly include “cell type that secrete sEVs” and “species that sEVs acting on”. Meanwhile, we further improve the comment about various effects produced by sEVs.

Reviewer 2 Report

Zhi Li and colleagues present a summary paper on an interesting topic. As an indication of the novelty of the topic, a simple search of the Pubmed database using the keywords in titles "exosome" AND "osteoarthritis" yields only 9 results, while 55 publications include these keywords in the abstract. The manuscript is well structured and logically organized, but I find some discrepancy between title, objective and content. The authors describe the global, and regulator, non-coding RNA-specific, effects of exosomes on major joint cell types. As a clarification, this should be mentioned in the introduction, because the available data on the effects of the protein nature of exosomes are limited. However, the dataset on the effects of miRNA, lcRNA, ciRNA components is growing rapidly. 

For this reason, I suggest a change in the title: "The dual role of exosomes in joint osteoarthritis: the global and non-coding regulator RNA molecule-based pathogenicity and therapeutic effect" would better fit the structure and content of the manuscript. 

Important improvements should be made before publication of the manuscript. Comments on content and form:

1. Subsections 3.1, 3.2, 3.3, 3.4 highlight the positive and negative exosome effects on chondrocytes, extracellular matrix, synovium and subchondral bone. The targets of these effects include inflammatory cytokines and several degradative enzymes (e.g. ADAMTS 4 and 5). However, data on NF-KB as a regulator of inflammation in chondrocytes, on the RANK-RANKL system in subchondral bone and only one case of subchondral osteoclasts are mentioned.

2. Neither in subsection 3.1 nor in chapter 4, tables 1, 2, 3, there is little or no mention of hypertrophic (one mention) and senescent chondrocytes, which are very important for osteoarthritis progression. The tables, miRNA, lcRNA, ciRNA effects should be supplemented with relevant data.

3. data on joint inflammation should also be presented to include RNA regulation of danger signals and transcription factors that regulate inflammation. 

4. the titles of subsections 3.1, 3.2, 3.3, 3.4 should be in capital letters: 'pathogenic exosomes', 'therapeutic exosomes'.

If these aspects are corrected, I agree with the publication of the work of Zhi Li et al. in Biomolecules.

Minor spelling issues should be corrected.

Author Response

Dear Editorial Board of Biomolecules,

We really appreciate the chance from Biomolecules for revising our manuscript. To address the questions from the editors and reviewers, we have carefully revised the ‘Introduction’, ‘Effects of sEVs on articular tissues in OA’, ‘Direct bearers of sEV effects on OA: Non-coding Regulatory RNA Molecule’ and ‘Limitations in the mechanism research for OA diagnosis and treatment’ sections accordingly.

Reviewer 2:

  1. As a clarification, this should be mentioned in the introduction, because the available data on the effects of the protein nature of exosomes are limited. However, the dataset on the effects of miRNA, lcRNA, ciRNA components is growing rapidly. For this reason, I suggest a change in the title: "The dual role of exosomes in joint osteoarthritis: the global and non-coding regulator RNA molecule-based pathogenicity and therapeutic effect" would better fit the structure and content of the manuscript.

We really appreciate your kind comment about our work. We totally agree that the available data on the effects of the protein nature of exosomes are limited while the data on non-coding regulator RNA is growing rapidly, and thank you very much for your valuable suggestion. We change the title: “The dual role of small extracellular vesicles in joint osteoar-thritis: Their global and non-coding regulatory RNA mole-cule-based pathogenic and therapeutic effects”, meanwhile delete “4.4 protein” section. We hope that there will be more reports on the role of sEV-proteins in OA in the future.

  1. Subsections 3.1, 3.2, 3.3, 3.4 highlight the positive and negative exosome effects on chondrocytes, extracellular matrix, synovium and subchondral bone. The targets of these effects include inflammatory cytokines and several degradative enzymes (e.g. ADAMTS 4 and 5). However, data on NF-KB as a regulator of inflammation in chondrocytes, on the RANK-RANKL system in subchondral bone and only one case of subchondral osteoclasts are mentioned.

This is a very important comment. We totally agree with the important role of NF-κB signaling pathway and RANK-RANKL system in OA, and add discussion in “4.1. MicroRNAs (miRNAs)” and “3.2.3. EFFECTS OF THERAPEUTIC SEVS ON SUBCHONDRAL BONE” sections.

  1. Neither in subsection 3.1 nor in chapter 4, tables 1, 2, 3, there is little or no mention of hypertrophic (one mention) and senescent chondrocytes, which are very important for osteoarthritis progression. The tables, miRNA, lcRNA, ciRNA effects should be supplemented with relevant data.

Thank you very much for your comment. We add discussion in “3.1.1. EFFECTS OF PATHOGENIC SEVS ON CHONDROCYTES” and “3.2.1. EFFECTS OF THERAPEUTIC SEVS ON CHONDROCYTES” sections. At the same time, we enrich the table.

  1. data on joint inflammation should also be presented to include RNA regulation of danger signals and transcription factors that regulate inflammation.

This is a very important comment. We enrich the table in “4. Direct bearers of sEV effects on OA” section and add specific changes in mRNA and protein expression caused by sEVs.

  1. the titles of subsections 3.1, 3.2, 3.3, 3.4 should be in capital letters: 'pathogenic exosomes', 'therapeutic exosomes'.

Thank you very much for your careful review. We have made formatting changes to the titles. However, because we improved article structure, the levels of " pathogenic exosomes " and "herapeutic exosomes" titles have changed, so we make modifications to the new corresponding levels of titles: “EFFECTS OF PATHOGENIC SEVS ON CHONDROCYTES”, “EFFECTS OF PATHOGENIC SEVS ON CARTILAGE EXTRACELLULAR MATRIX”……

Reviewer 3 Report

The work analyzes the dual effect of EVs in osteoarthritis. Authors underlined that EVs may either promote pathological progression or have a beneficial effect, depending on the cell source. The topic is of great interest and raises an important question in the field. However, a moderate editing of the English quality would be preferred and strongly suggested.

The quality of English language is not perfect and the text is therefore sometimes difficult to read and not very fluent. Editing of the text would be recommended in order to make the language more smooth.

Author Response

Dear Editorial Board of Biomolecules,

We really appreciate the chance from Biomolecules for revising our manuscript. To address the questions from the editors and reviewers, we have carefully revised the ‘Introduction’, ‘Effects of sEVs on articular tissues in OA’, ‘Direct bearers of sEV effects on OA: Non-coding Regulatory RNA Molecule’ and ‘Limitations in the mechanism research for OA diagnosis and treatment’ sections accordingly.

Reviewer 3:

  1. The work analyzes the dual effect of EVs in osteoarthritis. Authors underlined that EVs may either promote pathological progression or have a beneficial effect, depending on the cell source. The topic is of great interest and raises an important question in the field. However, a moderate editing of the English quality would be preferred and strongly suggested.

Thank you very much for your kind comment about our work. We have used professional English editing services to revise the manuscript, as shown below.

Round 2

Reviewer 1 Report

This review on regulatory RNA molecules delivered by extracellular vesicles in osteoarthritis provides a useful tool for researchers in the field. The authors made improvements in this version of the manuscript. The revisions answered my concerns. I have no more questions for the authors to answer.

Reviewer 2 Report

The authors provided significant improvements in the new version of the manuscript. I think that in this form, their work complies with the standards of Biomolecules'. 

Minor English editing may be necessary.